# Empirical Mode Decomposition Based Multi-Modal Activity Recognition

**DOI:** 10.3390/s20216055

**Published:** 2020-10-24

**Authors:** Lingyue Hu, Kailong Zhao, Xueling Zhou, Bingo Wing-Kuen Ling, Guozhao Liao

**Affiliations:** School of Information Engineering, Guangdong University of Technology, Guangzhou 511006, China; hulingyue1005@163.com (L.H.); iszhaokl@gmail.com (K.Z.); zzhouxueling@gmail.com (X.Z.); lgz13377589636@163.com (G.L.)

**Keywords:** activity recognition, multi-modal, empirical mode decomposition, random forest, electroencephalograms, image sequences, motion signals

## Abstract

This paper aims to develop an activity recognition algorithm to allow parents to monitor their children at home after school. A common method used to analyze electroencephalograms is to use infinite impulse response filters to decompose the electroencephalograms into various brain wave components. However, nonlinear phase distortions will be introduced by these filters. To address this issue, this paper applies empirical mode decomposition to decompose the electroencephalograms into various intrinsic mode functions and categorize them into four groups. In addition, common features used to analyze electroencephalograms are energy and entropy. However, because there are only two features, the available information is limited. To address this issue, this paper extracts 11 different physical quantities from each group of intrinsic mode functions, and these are employed as the features. Finally, this paper uses the random forest to perform activity recognition. It is worth noting that the conventional approach for performing activity recognition is based on a single type of signal, which limits the recognition performance. In this paper, a multi-modal system based on electroencephalograms, image sequences, and motion signals is used for activity recognition. The numerical simulation results show that the percentage accuracies based on three types of signal are higher than those based on two types of signal or the individual signals. This demonstrates the advantages of using the multi-modal approach for activity recognition. In addition, our proposed empirical mode decomposition-based method outperforms the conventional filtering-based method. This demonstrates the advantages of using the nonlinear and adaptive time frequency approach for activity recognition.

## 1. Introduction

Activity recognition plays an important role in many research areas. For example, activity recognition via electroencephalograms helps in the understanding of the working principles of the human brain. Activity recognition via motion signals also helps physical therapists to evaluate the effectiveness of rehabilitation. Moreover, activity recognition via image sequences can be used in security surveillance [1].

Recently, reading recognition using electroencephalograms was proposed [2]. First, the electroencephalograms are filtered by a bandpass filter to suppress noise and to reduce the unwanted movements of the subjects. Then, the electroencephalograms are decomposed into the β, α, θ, and δ waves using the short-time discrete Fourier transform [3,4]. The sums of the absolute discrete Fourier transform coefficients form the feature vectors and the k-nearest neighbor algorithm with *k* = 3 is used as the classifier.

Moreover, music preference analysis is performed via the electroencephalograms [5]. In particular, the sixth-order Butterworth bandpass filter is employed to decompose the electroencephalograms into the δ, θ, α, β, and γ waves. The sum of the magnitudes of the samples in the time frequency plane and the energy of the Hilbert Huang spectrum are employed as the features. The support vector machine, the k-nearest neighbor algorithm, the quadratic discriminant analysis, and the Mahalanobis distance-based discriminant analysis are used as the classifiers.

In addition, exergaming recognition via motion signals and relaxation analysis using electroencephalograms was proposed [6]. First, a long short-term memory neural network is used to classify the motion signals into various exercises, and the signal strengths of the electroencephalograms are used to calculate the meditation levels of the subjects.

Activity recognition via image sequences was also proposed [7]. In particular, the extended chain graph is employed to parameterize the joint probability distribution of a model to perform the learning.

However, the existing methods [2,5] employ infinite impulse response filters, such as the Butterworth filter, to extract the β, α, θ, γ, and δ waves of the electroencephalograms to perform brain wave analysis. Nevertheless, these filters introduce nonlinear phase distortions to the extracted waves. Hence, the classification accuracies based on these extracted waves are low. Furthermore, the existing methods only employ the sum of the magnitudes and the energy of the samples in the time frequency plane as features [2,5]. This limited number of features also contributes to the low classification accuracies.

To address the above difficulties, this paper proposes the use of empirical mode decomposition to decompose electroencephalograms into various intrinsic mode functions. These intrinsic mode functions are categorized into four groups. In addition, 11 different physical quantities are extracted from each group of intrinsic mode functions and employed as the features. Moreover, because different types of signals carry different information for performing activity recognition, a multi-modal approach using the electroencephalograms, the image sequences, and the motion signals is used for activity recognition. The outline of this paper is as follows. Section 2 presents the details of the multi-modal activity recognition. The computer numerical simulation results are shown in Section 3. Finally, a conclusion is drawn in Section 4.

## 2. Multi-Modal Activity Recognition

The objective of this paper is to perform activity recognition via three types of signals. Here, seven common activities are classified. These are: (1) watching television; (2) playing with toys; (3) eating; (4) playing electronic games; (5) performing online exercises; (6) reading/writing; and (7) drawing. The signals employed for activity recognition are electroencephalograms, image sequences, and motion signals.

### 2.1. Feature Extraction

#### 2.1.1. Features Extracted from the Electroencephalograms

The empirical mode decomposition assumes that a signal can be represented as the sum of a finite number of intrinsic mode functions. The intrinsic mode functions are obtained using the following procedures:

Step 1:Initialization: let $r_{0}\left(t\right)=x\left(t\right)$, i=1 and a threshold value equal to 0.3.Step 2:Let the *i*th intrinsic mode function be $c_{i}\left(t\right)$. This can be obtained as follows:(a)Initialization: let $d_{0}\left(t\right)=r_{i-1}\left(t\right)$, i=1 and j=1.
(b)Find all the maxima and minima of $d_{j-1}\left(t\right)$.
(c)Denote the upper envelope and the lower envelope of $d_{j-1}\left(t\right)$ as $e_{+}\left(t\right)$ and $e_{-}\left(t\right)$, respectively. Obtain $e_{+}\left(t\right)$ and $e_{-}\left(t\right)$ by interpolating the cubic spline function at the maxima and the minima of $d_{j-1}\left(t\right)$, respectively.
(d)Let the mean of the upper envelope and the lower envelope of $d_{j-1}\left(t\right)$ be m(t).
(e)Define $d_{j}\left(t\right)=d_{j-1}\left(t\right)-m\left(t\right)$.
(f)Compute $SD=\frac{\sum {\left|m\left(t\right)\right|}^{2}}{\sum {\left|d_{j-1}\left(t\right)\right|}^{2}}$. If *SD* is not greater than the given threshold, then set $c_{i}\left(t\right)=d_{j}\left(t\right)$. Otherwise, increment the value of j and go back to Step (b).


Step 3: Set $r_{i}\left(t\right)=r_{i-1}\left(t\right)-c_{i}\left(t\right)$. If $r_{i}\left(t\right)$ satisfies the properties of the intrinsic mode function or it is a monotonic function, then the decomposition is completed.

The details of the empirical mode decomposition can be found in [8,9,10]. Because a signal with more extrema will contain more high-frequency components, the intrinsic mode functions with the lower indices will be localized in the higher frequency bands. Hence, the empirical mode decomposition is a kind of time frequency analysis. Because of these desirable properties, this paper applies empirical mode decomposition to decompose the electroencephalograms into various intrinsic mode functions.

However, because the total number of intrinsic mode functions is determined automatically by the above algorithm, it is difficult to obtain the fixed length feature vectors to perform activity recognition. To tackle this difficulty, the intrinsic mode functions are grouped together. Because there are four to eight intrinsic mode functions for most of the electroencephalograms, the intrinsic mode functions are categorized into four groups. Let $I_{1}$, $I_{2}$, $I_{3}$, and $I_{4}$ be the sets of the first, second, third, and fourth groups of intrinsic mode functions, respectively.

If there are only four intrinsic mode functions obtained in the empirical mode decomposition, then each set of intrinsic mode functions contains one intrinsic mode function. That is, $c_{1}\left(t\right)\in I_{1}$, $c_{2}\left(t\right)\in I_{2}$, $c_{3}\left(t\right)\in I_{3}$ and $c_{4}\left(t\right)\in I_{4}$.

If there are only five intrinsic mode functions obtained in the empirical mode decomposition, then the third and fourth intrinsic mode functions are combined together as one group. That is, $c_{1}\left(t\right)\in I_{1}$, $c_{2}\left(t\right)\in I_{2}$, $c_{3}\left(t\right)+c_{4}\left(t\right)\in I_{3}$ and $c_{5}\left(t\right)\in I_{4}$. If there are only six intrinsic mode functions obtained in the empirical mode decomposition, then the second and third intrinsic mode functions are combined together as one group, and the fourth and fifth intrinsic mode functions are combined together as another group. That is, $c_{1}\left(t\right)\in I_{1}$, $c_{2}\left(t\right)+c_{3}\left(t\right)\in I_{2}$, $c_{4}\left(t\right)+c_{5}\left(t\right)\in I_{3}$ and $c_{6}\left(t\right)\in I_{4}$. If there are only seven intrinsic mode functions obtained in the empirical mode decomposition, then the first and second intrinsic mode functions are combined together as one group, the third and fourth intrinsic mode functions are combined together as another group, and the fifth and sixth intrinsic mode functions are combined together as another group. That is, $c_{1}\left(t\right)+c_{2}\left(t\right)\in I_{1}$, $c_{3}\left(t\right)+c_{4}\left(t\right)\in I_{2}$, $c_{5}\left(t\right)+c_{6}\left(t\right)\in I_{3}$ and $c_{7}\left(t\right)\in I_{4}$. If there are eight intrinsic mode functions obtained in the empirical mode decomposition, then the first and second intrinsic mode functions are combined together as one group, the third and fourth intrinsic mode functions are combined together as another group, the fifth and sixth intrinsic mode functions are combined together as another group, and the seventh and eighth intrinsic mode functions are combined together as another group. That is, $c_{1}\left(t\right)+c_{2}\left(t\right)\in I_{1}$, $c_{3}\left(t\right)+c_{4}\left(t\right)\in I_{2}$, $c_{5}\left(t\right)+c_{6}\left(t\right)\in I_{3}$ and $c_{7}\left(t\right)+c_{8}\left(t\right)\in I_{4}$.

Because the magnitudes of various brain waves for performing different activities are different, the magnitudes and the energies of various brain waves are usually employed as the features for activity recognition. Similar but more physical quantities are employed as the features in this paper. In particular, the entropy, mean, interquartile range, mean absolute deviation, range, variance, skewness, kurtosis, *L*_2_ norm, *L*_1_ norm, and *L*_∞_ norm of each group of intrinsic mode functions are computed and employed as the features [11,12]. Here, there are four groups of intrinsic mode functions for each electroencephalogram and there are 11 features extracted from each group of intrinsic mode functions. Hence, the lengths of each feature vector is 44.

#### 2.1.2. Features Extracted from the Image Sequences

Because different activities involve different objects, the objects are segmented from each image. Due to the movements of the subjects, the camera rotates and translates. As a result, the sizes of the same object in two consecutive images are different. To address this difficulty, because the discrete cosine transform can be used to resize the objects, the discrete cosine transform is first applied to the objects. Next, the matrices of the discrete cosine transform coefficients of the objects in two consecutive images are compared [13]. Then, the zeros are placed into the matrix of the discrete cosine transform coefficients corresponding to the smaller size of objects such that the size of the zero-filled matrix of the discrete cosine transform coefficients is the same as that of the matrix of the discrete cosine transform coefficients without zeros [14]. The zero-filled matrix or the matrix without zeros of the discrete cosine transform coefficients of the object in the ith image is denoted $D_{i}$.

It is worth noting that the rates of change of the objects in the image for different activities are different. For example, the rates of change of the objects in the image of the computer screen for playing electronic games are faster than those for performing the online exercises. This implies that the changes of the positions of the objects between two consecutive images can be employed as the features for activity recognition. Let the minimum x-coordinate, the maximum x-coordinate, the minimum y-coordinate, and the maximum y-coordinate of the object in the ith image be $x_{\min,i}$, $x_{\max,i}$, $y_{\min,i}$, and $y_{\max,i}$, respectively. The middle point of the x-coordinate and the middle point of the y-coordinate of the object in the ith image are defined as $x_{\mathrm{mean},i}$ and $y_{\mathrm{mean},i}$, respectively. In particular, $x_{\mathrm{mean},i}=\frac{x_{\max,i}+x_{\min,i}}{2}$ and $y_{\mathrm{mean},i}=\frac{y_{\max,i}+y_{\min,i}}{2}$. Here, $x_{\min,i+1}-x_{\min,i}$, $x_{\max,i+1}-x_{\max,i}$, $y_{\min,i+1}-y_{\min,i}$, $y_{\max,i+1}-y_{\max,i}$, $\left|x_{\mathrm{mean},i+1}\right|-\left|x_{\mathrm{mean},i}\right|$, $\left|y_{\mathrm{mean},i+1}\right|-\left|y_{\mathrm{mean},i}\right|$, and ${\left(\left|x_{\mathrm{mean},i+1}\right|-\left|x_{\mathrm{mean},i}\right|\right)}^{2}+{\left(\left|y_{\mathrm{mean},i+1}\right|-\left|y_{\mathrm{mean},i}\right|\right)}^{2}$ are employed as the features. In addition to using the features in the spatial domain, this paper also extracts the features based on the differences of the discrete cosine transform coefficients of the objects between two consecutive images. In particular, the mean, median, variance, skewness, and kurtosis of all of the coefficients in $D_{i+1}-D_{i}$ are also employed as the features. Obviously, the length of the feature vectors is 12.

#### 2.1.3. Features Extracted from the Motion Signals

It is worth noting that the positions and the angles of the camera are different for different activities. For example, the head is pointing forward during watching television, whereas it is pointing downward for reading/writing and drawing. Furthermore, the movements of the head are different for different activities. For example, the head moves more for eating. Hence, the mean and variance of the x-direction, the y-direction, and the z-direction of the motion signals are employed as the features [15,16]. Obviously, the length of the feature vectors is 6.

#### 2.1.4. Fusion of All the Features Together

The features extracted from each electroencephalogram, each image, and each motion signal are combined to form a feature vector. Here, the length of the feature vectors is 62.

### 2.2. Classification

A random forest is an extended variant of bagging. It consists of a collection of a large number of individual decision trees. However, it differs from bagging in the sense that each node variable is generated only from a handful of the randomly selected variables. Therefore, not only is the sample random, but the generation of each node variable (feature) is also random. Each tree in the random forest gives a class prediction and the class that votes the most becomes the prediction of the model [17]. The procedures for performing the random forest are summarized as follows and shown in Figure 1:

Step 1:If there are *N* samples, then these *N* samples are selected in a random sequence. Here, each sample is selected randomly at each time. That is, the algorithm selects another sample randomly after the previous sample is selected. These selected *N* samples form the decision nodes and are used to train a decision tree.Step 2:Suppose that each sample has *M* attributes; *m* attributes are selected randomly such that *m* << *M* is satisfied. Then, some strategies such as the information gain are adopted to evaluate these *m* attributes. Each node of the decision tree needs to split. One attribute is selected as the split attribute of the node.Step 3:During the formation of the decision tree, each node is split according to Step 2 until it can no longer be split.Step 4:Repeat Step 1 to Step 3 to establish a large number of decision trees. Thus, a random forest is formed.

From the above, it can be seen that different strategies, such as the information gain, can be adopted in the random forest. Hence, if an appropriate strategy is selected, then a high classification accuracy can be achieved. Moreover, due to the introduction of two sources of randomness, i.e., from the samples and from the features, the random forest does not easily suffer from the problem of overfitting. Furthermore, using the tree structure can help the model to address the issue of nonlinear data [18,19]. In addition, in the training process, the interaction among the features can be exploited and the importance of the features can be ranked accordingly.

Because of the above advantages, this paper adopts the random forest to extract the features and perform the classification. In particular, the random forest selects five of these 62 features and classifies the feature vectors into seven activities. Here, 30% of the overall data are employed for training and the remaining 70% are employed for testing. The total number of data points in the training and testing sets is summarized in Table 1. For simplicity, no cross-validation is performed.

### 2.3. Computational Complexity Analysis

It is important to investigate the required computational complexity of the algorithm. It is worth noting that the random forest is the module that requires the heaviest computational power. Let N be the total number of samples, M the total number of features, and D be the depth of the trees. When the classification and the regression tree (CART) grows, the values in all of the features of all samples are taken as the candidates for performing the splitting. The evaluation index, such as the information gain, gain ratio, or Gini coefficient, is calculated. Therefore, the required computational power for each layer of the random forest is O(N×M). Because there are D layers in the tree, the required computational power for the random forest is O(N×M×D). Furthermore, the spatial complexity of the random forest is O(N+M×Split×TreeNum), where *Split* is the average number of segmentation points for each feature and *TreeNum* is the total number of trees in the random forest. In the numerical simulation results, N values are chosen as 1336, 112, 1044, 1422, and 400 for volunteers 1 to 5, respectively, M is chosen as 62, and *TreeNum* is chosen as 100. In addition, *Split* is set to its default value of $10^{-7}$ and D is automatically determined. The processing time of the proposed method is about 2.094193 s.

## 3. Computer Numerical Simulation Results

Here, the full set of measurements was provided by five volunteers including two girls and three boys. The electroencephalograms were acquired by a single channel device and sampled at 512 Hz. Motion data was sampled at 31 Hz and the red green blue (RGB) images were taken at 0.1 Hz. The data acquisition for each volunteer took between 6 and 10 min, with most data acquisitions taking 10 min. A set of motion signals was taken randomly from both the training and the testing sets, and these signals were conducted by the first volunteer performing various activities. These motion signals are shown in Figure 2. It can be seen that the motion signals in the training set are consistent with those of the testing set.

Conventional electroencephalogram-based activity recognition applies various filters to the electroencephalograms to obtain various waves. In particular, the electroencephalograms are localized in the frequency band between 0.5 and 49 Hz. The frequency band of the δ wave is between 0.5 and 4 Hz, that of the θ wave is between 4 and 8 Hz, that of the α wave is between 8 and 12 Hz, that of the sensory motor rhythm (SMR) wave is between 12 and 14.99 Hz, that of the mid-β wave is between 15 and 19.99 Hz, that of the high-β wave is between 20 and 30 Hz, that of the low-β wave is between 12 and 19 Hz, that of the whole-β wave is between 12 and 30 Hz, and that of the γ wave is between 30 and 49 Hz. To extract these waves from the electroencephalograms, the fast Fourier transform approach is employed. That is, the fast Fourier transform coefficients of the electroencephalograms are computed and the coefficients outside the corresponding frequency bands are set to zero. Then, the inverse fast Fourier transform is computed to obtain the corresponding waves. It can be seen from the above that this approach does not introduce nonlinear phase distortion.

To investigate the effectiveness of applying the empirical mode decomposition to the electroencephalograms for activity recognition, Figure 3 plots the magnitude responses of the decomposed components of the electroencephalograms via both empirical mode decomposition and conventional filtering when the first and second volunteers perform various activities. Then, the physical quantities discussed in Section 2.1 are calculated for each wave and these physical quantities are employed as the features for performing the classification. To evaluate the performance of our proposed empirical mode decomposition-based method, it is compared to the above conventional filtering-based method. Here, the same set of physical quantities is computed for the comparison. It can be seen from the figure that the empirical mode decomposition can yield intrinsic mode functions with very narrow bandwidths. Hence, the features extracted from the intrinsic mode functions are more specific. As a result, the empirical mode decomposition approach can yield higher average classification accuracy.

To investigate the effects of various signals on the activity recognition, Figure 4 plots some of the features extracted from the motion signals when various volunteers perform various activities. Similarly, Figure 5 plots some of the features extracted from the image sequences when various volunteers perform various activities. Figure 6 plots some of the features extracted from the electroencephalograms when the first and second volunteers perform various activities. It can be seen from these figures that the features of various signals corresponding to different activities are localized in different regions in the feature space. This implies that the features of these signals are effective.

The percentage accuracy and the macro F1 score are used as the metrics to evaluate the performance of various methods. This is because these are the common criteria used in the classification problems. Table 2, Table 3, Table 4, Table 5 and Table 6 show the percentage accuracies and the macro F1 scores obtained by both our proposed empirical mode decomposition-based method and the conventional filtering-based method using the signals acquired from five different volunteers, respectively. It can be seen from the tables that the percentage accuracies based on three types of signal are higher than those using two types of signal. Furthermore, the percentage accuracies based on two types of signal are higher than those using the corresponding individual signals. Although this is not the case for all of the macro F1 scores, this is true for most of the cases. This demonstrates the advantages of using the multi-modal approach for activity recognition.

Moreover, using three types of signal, our proposed empirical mode decomposition-based method outperforms the conventional filtering-based method for the last four volunteers. Although this is not the case for the first volunteer, the difference is very small and can be ignored. To understand why there is an exception for the first volunteer, it can be seen from Figure 6 that the overlaps of the features among various activities in the feature space based on the empirical mode decomposition approach are larger than those based on the conventional filtering approach for the first volunteer. However, this is not the case for the second volunteer. This accounts for the exception. Overall, the obtained results demonstrate the advantages of using the nonlinear and adaptive time frequency approach for activity recognition.

## 4. Conclusions

This paper applies empirical mode decomposition to electroencephalograms to obtain the intrinsic mode functions localized in various frequency bands. The intrinsic mode functions are categorized into four groups. Then, 11 physical quantities are computed for each group of the intrinsic mode functions and used as features. Finally, the random forest is employed to perform multi-modal activity recognition. Numerical simulation results show that the percentage accuracies for activity recognition range between 78.21% and 96.90%. This demonstrates that the activities can be successfully recognized by our proposed algorithm. Furthermore, it can be seen that the percentage accuracies based on three types of signal are higher than those using two types of signal or individual signals. This demonstrates the success of using the multi-modal approach for activity recognition. Moreover, the numerical simulation results also show that the empirical mode decomposition-based method outperforms the conventional filtering-based method. This demonstrates the effectiveness of using the nonlinear adaptive approach to decompose the signal into various components for performing activity recognition.

## Figures and Tables

**Figure 1 sensors-20-06055-f001:**
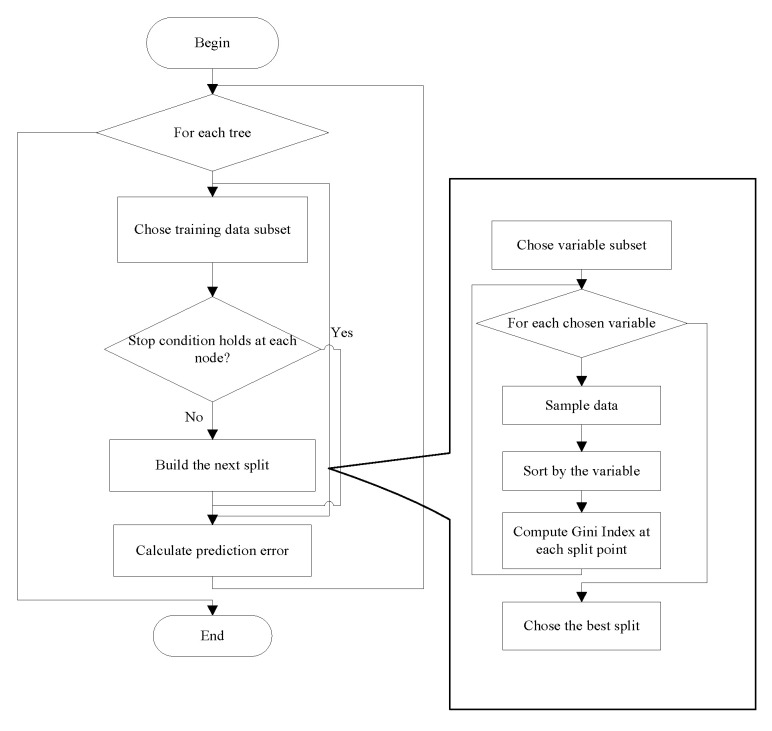
Flowchart showing the procedures of the random forest.

**Figure 2 sensors-20-06055-f002:**
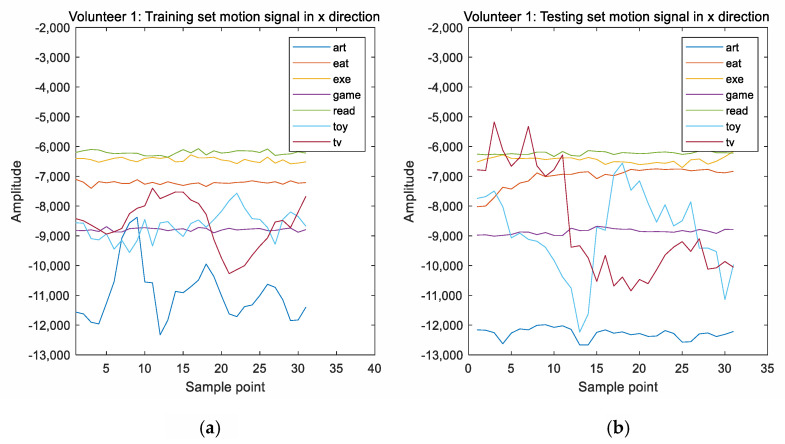
(**a**) The motion signal in the x-direction from the training set. (**b**) The motion signal in the x-direction from the testing set. (**c**) The motion signal in the y-direction from the training set. (**d**) The motion signal in the y-direction from the testing set. (**e**) The motion signal in the z-direction from the training set. (**f**) The motion signal in the z-direction from the testing set.

**Figure 3 sensors-20-06055-f003:**
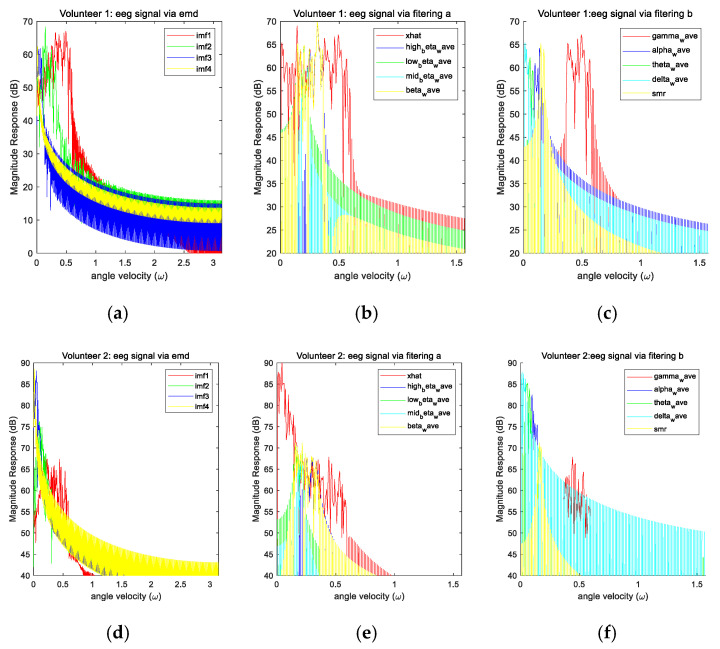
(**a**) The magnitude responses of the intrinsic mode functions of the electroencephalograms when the first volunteer performs various activities. (**b**) and (**c**) The magnitude responses of the filtered electroencephalograms when the first volunteer performs various activities. (**d**) The magnitude responses of the intrinsic mode functions of the electroencephalograms when the second volunteer performs various activities. (**e**) and (**f**) The magnitude responses of the filtered electroencephalograms when the second volunteer performs various activities.

**Figure 4 sensors-20-06055-f004:**
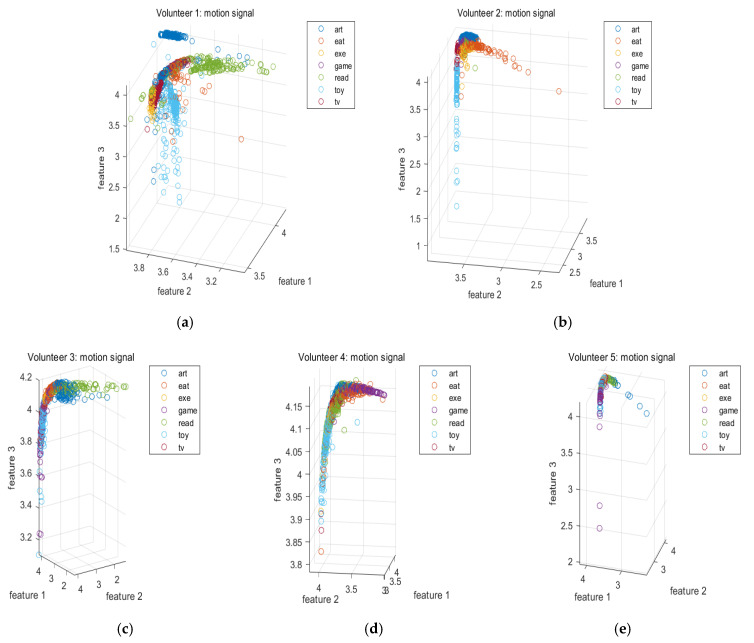
(**a**) Some of the features extracted from the motion signals when the first volunteer performs various activities. (**b**) Some of the features extracted from the motion signals when the second volunteer performs various activities. (**c**) Some of the features extracted from the motion signals when the third volunteer performs various activities. (**d**) Some of the features extracted from the motion signals when the fourth volunteer performs various activities. (**e**) Some of the features extracted from the motion signals when the fifth volunteer performs various activities.

**Figure 5 sensors-20-06055-f005:**
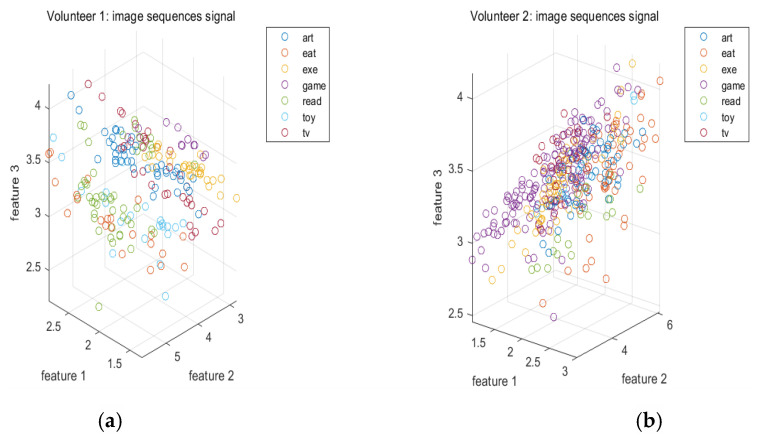
(**a**) Some of the features extracted from the image sequences when the first volunteer performs various activities. (**b**) Some of the features extracted from the image sequences when the second volunteer performs various activities. (**c**) Some of the features extracted from the image sequences when the third volunteer performs various activities. (**d**) Some of the features extracted from the image sequences when the fourth volunteer performs various activities. (**e**) Some of the features extracted from the image sequences when the fifth volunteer performs various activities.

**Figure 6 sensors-20-06055-f006:**
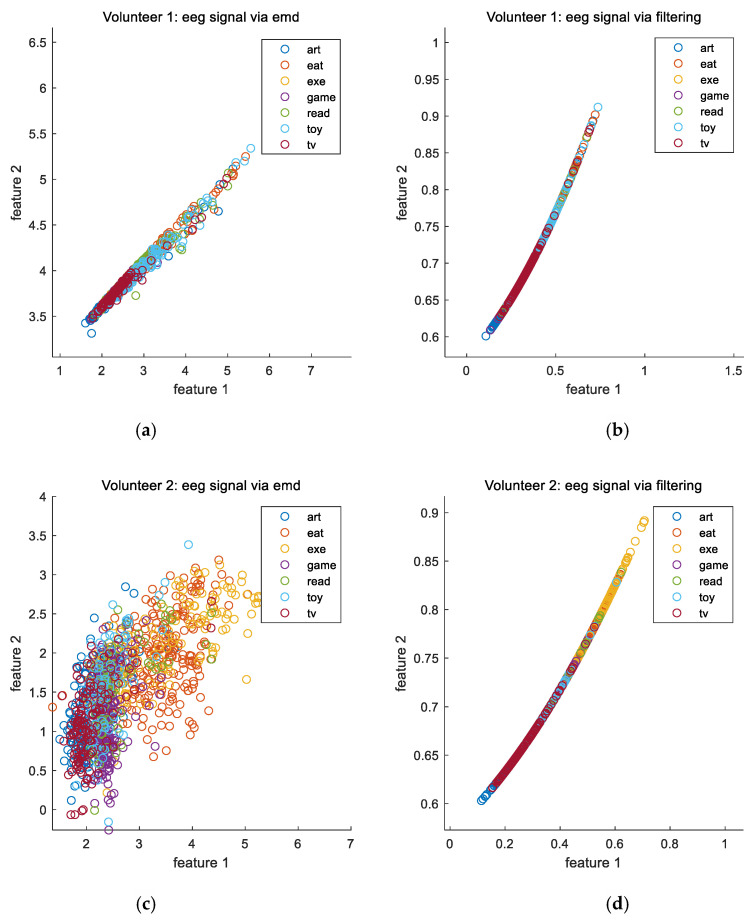
(**a**) Some of the features extracted from the electroencephalograms via the empirical mode decomposition approach when the first volunteer performs various activities. (**b**) Some of the features extracted from the electroencephalograms via the conventional filtering approach when the first volunteer performs various activities. (**c**) Some of the features extracted from the electroencephalograms via the empirical mode decomposition approach when the second volunteer performs various activities. (**d**) Some of the features extracted from the electroencephalograms via the conventional filtering approach when the second volunteer performs various activities.

**Table 1 sensors-20-06055-t001:** The total number of data points in the training and testing sets.

Volunteer Identity Number	1	2	3	4	5
Total number of data points in both the training set and the test set	1336	1122	1044	1422	400
Total number of data points in the training set	400	336	313	426	120
Total number of data points in the test set	936	786	731	996	280

**Table 2 sensors-20-06055-t002:** The percentage accuracies and the macro F1 scores obtained by our proposed empirical mode decomposition-based method and the conventional filtering-based method using the signals acquired from the first volunteer.

	The Percentage Accuracies and the Macro F1 Scores Obtained by Our Proposed Empirical Mode Decomposition-Based Method		The Percentage Accuracies and the Macro F1 Scores Obtained by the Conventional Filtering-Based Method	
	Percentage Accuracies	Macro F1 Scores	Percentage Accuracies	Macro F1 Scores
The results based on the motion signals, the electroencephalograms, and the image sequences	0.9690	0.8923	0.9733	0.8708
The results based on the motion signals and the image sequences	0.9466	0.8916	0.9466	0.8916
The results based on the electroencephalograms and the image sequences	0.9423	0.8172	0.9658	0.7858
The results based on the motion signals and the electroencephalograms	0.8921	0.7820	0.8953	0.8289
The results based on the image sequences	0.8590	0.8100	0.8590	0.8100
The results based on the electroencephalograms	0.2874	0.4081	0.4605	0.4765
The results based on the motion signals	0.8771	0.8171	0.8771	0.8171

**Table 3 sensors-20-06055-t003:** The percentage accuracies and the macro F1 scores obtained by our proposed empirical mode decomposition-based method and the conventional filtering-based method using the signals acquired from the second volunteer.

	The Percentage Accuracies and the Macro F1 Scores Obtained by Our Proposed Empirical Mode Decomposition-Based Method		The Percentage Accuracies and the Macro F1 Scores Obtained by the Conventional Filtering-Based Method	
	Percentage Accuracies	Macro F1 Scores	Percentage Accuracies	Macro F1 Scores
The results based on the motion signals, the electroencephalograms, and the image sequences	0.9326	0.8343	0.9237	0.8337
The results based on the motion signals and the image sequences	0.9186	0.8289	0.9186	0.8289
The results based on the electroencephalograms and the image sequences	0.8779	0.7798	0.8677	0.7313
The results based on the motion signals and the electroencephalograms	0.8384	0.7896	0.8397	0.7626
The results based on the image sequences	0.8410	0.7813	0.8410	0.7613
The results based on the electroencephalograms	0.4593	0.4177	0.5394	0.4859
The results based on the motion signals	0.8079	0.7638	0.8079	0.7438

**Table 4 sensors-20-06055-t004:** The percentage accuracies and the macro F1 scores obtained by our proposed empirical mode decomposition-based method and the conventional filtering-based method using the signals acquired from the third volunteer.

	The Percentage Accuracies and the Macro F1 Scores Obtained by Our Proposed Empirical Mode Decomposition-Based Method		The Percentage Accuracies and the Macro F1 Scores Obtained by the Conventional Filtering-Based Method	
	Percentage Accuracies	Macro F1 Scores	Percentage Accuracies	Macro F1 Scores
The results based on the motion signals, the electroencephalograms, and the image sequences	0.9384	0.8492	0.8960	0.8406
The results based on the motion signals and the image sequences	0.8782	0.8340	0.8782	0.8340
The results based on the electroencephalograms and the image sequences	0.8892	0.7349	0.8782	0.7586
The results based on the motion signals and the electroencephalograms	0.7373	0.7214	0.7442	0.7309
The results based on the image sequences	0.8536	0.7456	0.8536	0.7456
The results based on the electroencephalograms	0.3912	0.3965	0.3666	0.4489
The results based on the motion signals	0.6977	0.6709	0.6977	0.6709

**Table 5 sensors-20-06055-t005:** The percentage accuracies and the macro F1 scores obtained by our proposed empirical mode decomposition-based method and the conventional filtering-based method using the signals acquired from the fourth volunteer.

	The Percentage Accuracies and the Macro F1 Scores Obtained by Our Proposed Empirical Mode Decomposition-Based Method		The Percentage Accuracies and the Macro F1 Scores Obtained by the Conventional Filtering-Based Method	
	Percentage Accuracies	Macro F1 Scores	Percentage Accuracies	Macro F1 Scores
The results based on the motion signals, the electroencephalograms, and the image sequences	0.8494	0.8532	0.8464	0.7864
The results based on the motion signals and the image sequences	0.8394	0.8085	0.8394	0.8085
The results based on the electroencephalograms and the image sequences	0.8092	0.6463	0.8283	0.6232
The results based on the motion signals and the electroencephalograms	0.7028	0.6592	0.6687	0.6822
The results based on the image sequences	0.7892	0.6213	0.7892	0.6213
The results based on the electroencephalograms	0.3936	0.3833	0.3353	0.3660
The results based on the motion signals	0.6295	0.5848	0.6295	0.5848

**Table 6 sensors-20-06055-t006:** The percentage accuracies obtained by our proposed empirical mode decomposition-based method and the conventional filtering-based method using the signals acquired from the fifth volunteer.

	The Percentage Accuracies and the Macro F1 Scores Obtained by Our Proposed Empirical Mode Decomposition-Based Method		The Percentage Accuracies and the Macro F1 Scores Obtained by the Conventional Filtering-Based Method	
	Percentage Accuracies	Macro F1 Scores	Percentage Accuracies	Macro F1 Scores
The results based on the motion signals, the electroencephalograms, and the image sequences	0.7821	0.6961	0.7750	0.6317
The results based on the motion signals and the image sequences	0.7464	0.6904	0.7464	0.6904
The results based on the electroencephalograms and the image sequences	0.7643	0.6321	0.6964	0.5865
The results based on the motion signals and the electroencephalograms	0.5107	0.5066	0.4786	0.4839
The results based on the image sequences	0.6786	0.6656	0.6786	0.6656
The results based on the electroencephalograms	0.1857	0.2292	0.3107	0.2523
The results based on the motion signals	0.4429	0.4693	0.4429	0.4693

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
