# Peer review of "Empirical Mode Decomposition Based Multi-Modal Activity Recognition"

_sensors, 2020, doi:10.3390/s20216055_

Round 1

Reviewer 1 Report

In Figures 2(c) and 2(d), there are 7 legends but 6 curves. It seems that the art curves are missed. Please, add the remaining curves.

The authors addressed my concerns. After modifying the mentioned figures, the manuscript can be published.

Reviewer 2 Report

Unique work for which the revision has provided major improvements in clarity and purpose, and can be of interest to the journal audience on machine learning applications.

Author Response

This manuscript is a resubmission of an earlier submission. The following is a list of the peer review reports and author responses from that submission.

Round 1

Reviewer 1 Report

The proposed paper provides a unique and potentially informative technique for assessing multi-modal activity recognition, but needs significant rework to accomplish that objective.

A recommendation is the abstract be re-written to indicate the purpose of the work, reciting more than the details of the methods and the advantages, but also to place the work in context.  As is, the abstract suggests a signal analysis effort with no motivation. This is followed by an Introduction that switches between topics with no transition. The first paragraph provides “broad sweeping” statements that have no evidence to support them and are controversial, and it is a “reach” to see how the first paragraph justifies the signal analysis effort.

Line 32 - The idea that “ there are many intelligent home systems for monitoring the home situations, these 32 systems do not generate the written reports” is not clearly supported by Reference 1.  The entire paragraph needs to be reconsidered in the context of the motivation for the paper.

Line 36 “Since the activities done by a person are controlled by his/her electroencephalograms, the 36 electroencephalograms are employed for performing the activity recognition” is for epileptic seizure detection and it is challenging for the review to understand electroencephalograms become the exclusive basis for monitoring home activities of children.  Yes, indeed, several features are extracted, but it needs more of a compelling argument to support the objectives outline in Lines 61-67.

Lines 70-79 represent one sentence and there are so many concepts buried in this sentence it is very difficult for the reader to extract meaning.

Line 93 and the associated table need to be restructured so the reader can establish the significant of the coefficient.

Line 119 suggests a causal relation, joining two sentences with “…Hence, “ and the basis for feature extraction not evident.  The message may be there, but it is lost on the reviewer.  The subsequent Line 121-125 present common concepts and their inclusion is not helpful tot the reader.

Lines 143-153 would be more helpful in the form of a flowchart.

Line 155: Again, the argument does not stand on it’s merit; value is not self-evident: “From the above, it can be seen that the random forest adopts the integrated algorithm. Hence, it 155 can achieve a higher accuracy than most of the individual algorithms.”

Generally the analytics of the paper are sound, but the motivation and ability to validate the multi-mode system.  The results section appear to have been successful with the subjects tested, but the absence of detail on modes and actual subject activity is not clear to the reader.  The message is lost in the frequency decomposition study that are the key results of the paper.

The conclusion does not circle back to indicate that the objectives of the paper were met.

Reviewer 2 Report

In this manuscript, the authors utilized three different signals (electroencephalograms, RGB images, and motion signals) and the empirical mode decomposition technique to implement an activity recognition task for children's activities at home. From my point of view, the work should be improved in order to attain enough qualifications to be published in Sensors. I have the following concerns:

1) A figure should be added to show some examples of your data points in the training and the test sets (some examples of images and signals).

2) The number of data points in the training and the test sets should be mentioned.

3) The computational complexity and the processing time of the proposed method should be explained.

4) Providing just accuracy as the metric is not enough. I encourage you to present the F1-score too.

5) For Table 1, you should explain why the conventional filtering based method performs better than your approach for the first person. Which characteristics of this scenario cause inferior results for your method?

6) I recommend citing the following research papers for activity recognition:

a) Ahmadi, M.N.; Pavey, T.G.; Trost, S.G. Machine Learning Models for Classifying Physical Activity in Free-Living Preschool Children. Sensors, 2020, 20, 4364. 

b) Ni, Q.; Fan, Z.; Zhang, L.; Nugent, C.D.; Cleland, I.; Zhang, Y.; Zhou, N. Leveraging Wearable Sensors for Human Daily Activity Recognition with Stacked Denoising Autoencoders. Sensors, 2020, 20, 5114. 

Also, a filtering technique, as proposed in feature adaptive filtering, can be utilized to extract the decomposing components in different frequency bands of the signal. Thus, I suggest citing the following paper:

c) H. Yazdanpanah, P. S. R. Diniz and M. V. S. Lima, "Feature Adaptive Filtering: Exploiting Hidden Sparsity," in IEEE Transactions on Circuits and Systems I: Regular Papers, vol. 67, no. 7, pp. 2358-2371, July 2020.

7) Finally, you should edit the English style of the manuscript carefully. There are various typos; some examples are:

Line 77: components Here --> components. Here (the point is forgotten)

Lines 110 and 111: unzero --> nonzero

Line 142: model 1616 --> model 16